# *Vaccinium* Species (Ericaceae): Phytochemistry and Biological Properties of Medicinal Plants

**DOI:** 10.3390/molecules28041533

**Published:** 2023-02-05

**Authors:** Gheorghe Adrian Martău, Teleky Bernadette-Emőke, Răzvan Odocheanu, Dacian Andrei Soporan, Mihai Bochiș, Elemer Simon, Dan Cristian Vodnar

**Affiliations:** 1Faculty of Food Science and Technology, University of Agricultural Sciences and Veterinary Medicine Cluj-Napoca, Calea Mănăştur 3-5, 400372 Cluj-Napoca, Romania; 2Institute of Life Sciences, University of Agricultural Sciences and Veterinary Medicine Cluj-Napoca, Calea Mănăştur 3-5, 400372 Cluj-Napoca, Romania; 3CENCIRA Agrofood Research and Innovation Centre, Ion Meșter 6, 400650 Cluj-Napoca, Romania

**Keywords:** *Vaccinium uliginosum* L., *Ericaceae*, herbal medicine, bioactivity, in vitro and in vivo analyses, phytochemical analysis, drug discovery, traditional medicine

## Abstract

The *Vaccinium* L. (Ericaceae) genus consists of a globally widespread and diverse genus of around 4250 species, of which the most valuable is the *Vaccinioidae* subfamily. The current review focuses on the distribution, history, bioactive compounds, and health-related effects of three species: cranberry, blueberry, and huckleberry. Several studies highlight that the consumption of *Vaccinium* spp. presents numerous beneficial health-related outcomes, including antioxidant, antimicrobial, anti-inflammatory, and protective effects against diabetes, obesity, cancer, neurodegenerative diseases and cardiovascular disorders. These plants’ prevalence and commercial value have enhanced in the past several years; thus, the generated by-products have also increased. Consequently, the identified phenolic compounds found in the discarded leaves of these plants are also presented, and their impact on health and economic value is discussed. The main bioactive compounds identified in this genus belong to anthocyanins (cyanidin, malvidin, and delphinidin), flavonoids (quercetin, isoquercetin, and astragalin), phenolic acids (gallic, *p*-Coumaric, cinnamic, syringic, ferulic, and caffeic acids), and iridoids.

## 1. Introduction

Fruit production has been growing steadily in the last few decades based on the data reported by the United Nations Food and Agriculture Organization [1]. On the other hand, as the population is also increasing, and following recent trends regarding consumer preferences for organic, good-quality food products and healthy everyday living, the generated fruits are insufficient to provide the best quality of food globally [2,3]. Additionally, as fruit production increases, the generation of large quantities of by-products is inevitable [4,5,6]. Integrating these by-products into functional foods presents a promising alternative to reduce the generated waste and increase several foodstuffs’ nutritional quality [7,8,9,10,11].

Belonging to the Ericaceae (*Rhododendron*) genus, the *Vaccinium* species comprise a morphologically diverse genus of 4250 species (33 types), divided into 9 subfamilies and 125 genera prevalent across Europe, South East and Central Africa, Asia, and North and Central America [12,13,14]. The wild species of this genus that are specific to Europe are *V. myrtillus* L. (bilberry), *V. vitis-idaea* L. (lingonberry), *V. oxycoccus* L. (cranberry), and *V. uliginosum* L. (bog bilberry) [15]. The most cultivated species with the highest economic value belong to the subfamily of *Vaccinioidae*, and are particularly represented by cranberry, blueberry, huckleberry, and bilberry [16]. Due to its various health-related benefits, its consumption is in constant growth, being efficient in relation to diabetes, obesity, cardiovascular and neurodegenerative diseases, atherosclerosis, rheumatoid arthritis, and cancer [17,18]. Based on several studies, the leading health benefits are accredited to their antioxidant, antimicrobial, and detoxifying effects on the organism [19,20,21]. Additionally, it is considered that these plants, particularly those with high polyphenol quantity such as anthocyanins, are able to hinder and initiate apoptosis within cancer cells [22].

The importance of polyphenols from plants has been acknowledged in the last few years, mainly due to them being an essential element for the human microbiota. As they are characterized as prebiotics, “a substrate that is selectively utilized by host microorganisms conferring a health benefit,” they bestow protective properties on the cardiovascular, immune, gastrointestinal, and central nervous systems [23,24,25]. The principal aspects of their interest are related to the current pandemic, which has highlighted the importance of a strong immune system [26] and the efficiency of food packaging, such as active packaging enriched with bioactive compounds [27,28,29,30].

*Vaccinium* is a common and widespread genus of shrubs or dwarf shrubs in the heath family (*Ericaceae*). Humans eat the fruits of many species, and some are of commercial importance, including cranberry, blueberry, bilberry (whortleberry), lingonberry (cowberry), and huckleberry. These species, including their flowers, leaves, and fruits, are widely applied in traditional medicine [31]. The content of phenolic compounds in these plants is considerably affected by the level of development, the localization of the plant, and by genetic causes. An important element is the amount of sunlight, which greatly influences the quantity of phenolic compounds [32]. Similar to other types of plants, the fruits are the parts most often consumed, while the other parts, such as leaves and stems, are discarded [33,34,35].

The current review aims to describe the current knowledge regarding the species cranberry, blueberry, and huckleberry belonging to the *Vaccinium* genus; their fruits, leaves, and stems; and their phytochemistry and biological properties, with specific attention paid to their health-related effects and economic value.

## 2. Blueberry/Bilberry (*V. myrtillus*/*V*. sect. *cyanococcus*)

### 2.1. History, Nomenclature, and Production

Bilberry (*V. myrtillus*) is a perennial low-growing shrub which can grow up to 35–60 centimetres in height. It grows in acidic soils [36], organic forest soils, mountainous mineral heaths, and old peat bogs in central and northern parts of Europe [37]. The shrubs can also be found in parts of Asia and North America, where it is located under the name of blueberry (*V. corymbosum* L.). Bilberry is usually known as European blueberry or whortleberry [36]. *V. myrtillus* is a diploid shrub [15] which blooms from April through June. It produces spheroidal blue or black intensely coloured fruits that can grow up to 5–9 millimetres in diameter, with many seeds, that mature from July to September. The pulp has the shade of the peel, and the taste is sweet with astringent notes. An uncommon albino form is characterized by greenish-white fruits, where this appearance is caused by the overthrow of genes that code the anthocyanin synthesis [38]. Bilberry is generally known as a “super food”, thanks to its abundance in healthful compounds and its employment in human diet from ancient times. The first written shreds of evidence of decoction preparations from dried bilberry fruits to apply in folk medicine date from the 18th century [36].

The reputation of bilberry can be attributed to its balanced sweet and sour flavour and rich nutritional value. Bilberries collected from wild areas are consumed and commercially available mostly as berries (fresh, dried, and frozen) and in the form of canned industrial products, including jams, juices, and concentrates. The fruits are utilized, in combination with other products, to produce syrups, pies, tarts, and beverages. Leaves are generally used in order to produce decoctions [36,39]. The expanding demand for a berry-rich diet has led to an expansion in the consumption and management of two *Vaccinium* genus berries. Those breeds are the *V. myrtillus*, or wild berries, and *V. corymbosum*, or cultivated berries [40]. Over the last five years, the production of blueberry has mainly been focused in North America [41], accounting for over 80% of world production [42], and with a production that reached over 1 million (M) tons in 2021 [40,43]. The U.S. registered a 106% increase in blueberry production from 2007 to 2016 [42]. According to FAOSTAT, in 2020, a total cultivated blueberry area of 126.144 ha was recorded globally; a significant difference compared to 1980, when the cultivated area was 24.460 ha. Previously discussed data indicate an increase of approximately 303% in the last 20 years.

Blueberry production and consumption are predicted to go through the same interest and constant growth, enlarging by 22%; more accurately, over 1 M tons by 2030 [1]. In addition, considering the last 20 years of blueberry production and using Excel 2022 and FORECAST function, we predicted blueberry production until 2030 based on existing values that follow a seasonal trend. A confidence interval was estimated using interval values from 2000–2020. The interval was defined by its lower and upper bounds from 2020 until 2030. Seasonality was considered to be one decade. The confidence interval is expressed as a percentage (95%). The percentage reflects the confidence level. According to data analysis from FAO data, the blueberry forecast may reach a value of 1.08 ± 0.26 M tons by 2030. Figure 1 shows a constant forecast and trendline that will increase in the following years.

### 2.2. Blueberry Constituents

Bilberries incorporate high amounts of sugars (68.5–205.3 g/kg), organic acids (5.7–14.2 g/kg), and a mixture of phenolic compounds with multiple biological effects. Anti-carcinogenic, antimicrobial, and anti-inflammatory activities were indicated by in vitro studies performed on bilberries [39]. In terms of proximate composition, fresh bilberry fruits contain 84% water, 9.7% carbohydrates, 0.6% proteins, 0.4% fats, and 3–3.5% fibres, and their energetic value is estimated to be about 192 kJ/100 g [44,45].

Phenolic compounds represent the primary group of phytochemicals found in berries. They acquire one or more aromatic rings with hydroxyl groups, and their construction may range from a simple phenolic molecule to a complex high-molecular-mass polymer, as can be seen in Table 1, adapted from [32,38,46]. Many factors can determine the content of phenolic compounds in the aerial parts of berry plants; these can be genotype, growing conditions, ripeness, and storage conditions. Researchers demonstrated that the fruits of berry plants grown in a cold climate with a short vegetation season are characterized by a superior content of phenolic compounds rather than the same varieties that are grown in a moderate climate. Phenolic compounds can be used to determine authenticity, since genotype profoundly impacts the qualitative and quantitative composition of these compounds in berries [47].

#### 2.2.1. Anthocyanins

Anthocyanins are the primary components isolated and identified in berries (reaching up to 0.1–0.25% of the fresh fruits) and leaves, along with diverse, active constituents, including: resveratrol, flavonols (quercetin, catechins), phenolic acids, ellagitannins, and iridoids [36]. The fruits are defined by the existence of a complex mixture of over 20 different anthocyanins, each comprised of sugar(s) and an aglycone. Glucose and rhamnose are the most frequent sugar moieties hooked to the aglycone, but galactose, xylose, rutinose, arabinose and other sugars can also be found in that linkage. The anthocyanidins present a significant skeleton of the 2–phenyl–1–benzopyrylium cation. These form an entire group of phenolic substances characterized as plant secondary metabolites that determine the intense colouration of distinctive plant organs, mainly flowers, fruits, and the products obtained from them. Anthocyanins represent phenolic secondary metabolites associated with the large family of phenylchroman derivatives (flavonoid family). They are glycosides of anthocyanidins, of which the most significant in the plants are: petunidin, malvidin, peonidin, pelargonidin, delphinidin, and cyanidin, in a distribution of 7, 7, 12, 12, 12, and 50%, respectively [47,48].

Bilberries (*V. myrtillus* L.) represent the richest natural source of anthocyanins and possess the highest content of them (300–700 mg/100 g of fresh fruit), but the content may vary depending on the degree of ripeness, edaphoclimatic conditions, and cultivars. These components belonging to the flavonoid family confer bilberries their blue/black colour and high antioxidant properties [15]. *V. myrtillus* contains primarily delphinidin and cyanidin in a 1:1 proportion, followed by petunidin, peonidin, and malvidin, with a discrepancy in the substitution pattern of ring B. At least three distinctive sugars (β–D–glucose, β–D–galactose, and α–L–arabinose) are glycosidically associated with the C3 position. Inconsequential compounds are associated with the groups of flavan-3-ols and their respective glycosides, iridoids, phenolic acids, terpenes, organic acids, stilbenes, and pectins. Additionally, 1% of tannins (generally proanthocyanidins (PACs) and fewer amounts of ellagitannins) are found [36,38,48].

*V. myrtillus* possesses the most intensely coloured berries thanks to the fact that both peel and pulp contain a massive amount of anthocyanidins (up to 2% of the fresh mass in peels). Anthocyanins constitute about 90% of the total phenolic compounds in bilberry fruits. These fruits contain distinctive cyanidin- and delphinidin–3-*O*–sambubiosides, wherein the saccharide moiety is glucose (2 → 1) xylose. The isolation of 3-*O*–methyl anthocyanidins in large quantities from bilberry fruits is also feasible [38].

Anthocyanins from bilberry extracts are so essential that they have been regulated at quantities of 32 and 40% of anthocyanidins (aglycon of anthocyanins), as reported by the 8th edition of European Pharmacopoeia (2014) [49]. Furthermore, the glycosylated form of cyaniding and delphinidin accounts for 60% of the total amount of anthocyanins. Two studies delineated the berry anthocyanin metabolism. They stated that fifteen minutes following the ingestion of bilberry anthocyanins, there was an increase in the plasma level of the particles from 2.5 mg/L (400 mg/kg per os) to 50 mg/L (100 mg/kg for ethanol extract), and afterwards, a significant decline was seen. The primary anthocyanins in the plasma were malvidin–3–glucoside and malvidin–3–galactoside. Furthermore, anthocyanidins are characterized by a specific organotropism, particularly prevalent in the liver, kidney, testes, and lungs, but absent in the brain, heart, fat, muscle, and eyes [2]. Regarding its health effects, anthocyanins are described as a bioactive key capable of many reported benefits [50].

#### 2.2.2. Flavonols

From the flavonol point of view, quercetin is the principal flavonol of bilberry fruits, representing more than 50% of the total flavonoid content. The second most plentiful flavonol found in fruits is myricetin. Other flavonols found in low levels are: laricitrin, syringetin, and isorhamnetin. Kaempferol, another component of the flavonols family, is found only in trace amounts in fruits but in bountiful quantities in leaves. In *V. myrtillus*, flavonols develop primarily in glycosylated form, and studies showed diverse hexosides, pentosides, and glucuronides. The main glycosides found in bilberry fruits are rhamnosides and glucuronides. Furthermore, phlorizin, a glucoside of a dihydrochalcone phloretin, was found in these fruits, but the aglycone itself was not identified [38].

#### 2.2.3. Tannins, Procyanidins, and Organic Acids

All the plants of the *Vaccinium* genus represent one of the prosperous sources of tannins of both hydrolyzable and condensed types [51]. A B–type dimer of catechin was found in fruits, but in both leaves and fruits, epicatechin and catechin are found in abundance. Epigallocatechin and gallocatechin can be found mainly in leaves. Both fruits and leaves of bilberry (*V. myrtillus*) are abundant in procyanidin in diversified dimers and trimers. The universal B–type procyanidins are characterized by a single connection between structural units of catechins, while the rare A–type procyanidins by a double bond. The A–type linkage is dominant over the B–type in bilberries, which is abnormal in other plant species. *V. myrtillus* leaves and fruits represent a rich source of organic acids and their derivatives. Past studies showed that most of the total phenolic compounds are found in the leaves, and in the fruits, they are overshadowed by anthocyanins. The simple organic acids present in bilberry fruits are shikimic, citric, malic, and quinic. The composition of phenolic and hydroxycinnamic acids is dominated by p-coumaric, caffeic, and ferulic acids, followed by gallic, ellagic and syringic acids. In the fruits, vanillic acid, salicylic acid, and dihydroxybenzoic acid can also be found, but in a smaller amount. The organic acids mentioned above can be found either free, etherified, or esterified. The etherification process usually occurs with various saccharide fractions, and they can develop esters with each other or with different phenolic compounds. The most distinguished of these by-products are chlorogenic and neochlorogenic acids, which regularly make up the bulk of the total phenolic acids of bilberry fruits and leaves [38].

### 2.3. Antioxidant Effect

Attention to phenolic compounds has grown recently, mainly because they are exceptional antioxidants [52]. The consumption of antioxidants has proven very efficient in avoiding cancer, cardiovascular diseases, osteoporosis, diabetes, obesity, and skin ageing. The fields where the antioxidant properties of plant phenolic compounds are relevant are food (inhibition of lipid oxidation), physiology (protection against oxidative stress), and cosmetology. The antioxidant characteristics of polyphenols have been linked to their capacity to act as reducing agents. They reverse the UV filter and can connect with metal ions and proteins. Mainly, phenolic compounds produce antioxidant activity by precisely decreasing reactive oxygen species (ROS), restraining the enzymes that are participating in oxidative stress, associating metal ions engaged for the creation of ROS, and strengthening endogenous antioxidant defence systems. Dróżdż, Šėžienė, and Pyrzynska concluded that bilberry fruits displayed high antioxidant activity as suitable electron donors and their extracts were capable of lowering copper (II)-neocuproine chelate, as well as extinguishing 2,2–diphenyl–1–picrylhydrazyl radicals (DPPH) [32,44].

Principally, phenolic compounds employ their antioxidant activity by directly scavenging ROS, restraining the enzymes participating in oxidative stress, reconstructing of other antioxidants (α–tocopherol), chelating metal ions involved in ROS production, and incentivising the endogenous antioxidant defence system. Partial reduction of oxygen generates ROS, which develops radical oxygen species, such as anion superoxide, hydroxyl radical, nitric oxide, oxyl, and peroxyl radicals (peroxidation process of lipids). The most notorious approaches for the determination of the antioxidant activity of phenolic extracts are the Folin–Ciocalteu method, DPPH (2,2–diphenyl–1–picrylhydrazyl) radical scavenging method, oxygen radical absorbance capacity (ORAC), ferric-ion-reducing antioxidant power (FRAP), and Trolox, which involves equivalent antioxidant capacity (TEAC) and cupric ion reducing antioxidant capacity (CUPRAC). The Folin–Ciocalteu method uses gallic acid to access the total phenolic concept (TPC) from the bilberry extract. Additionally, it evaluates the ability of a bilberry sample to diminish the transition of the metal ions, similar to the complex between sodium phosphomolybdate and phosphotungstate. The DPPH test is based on the competence of lowering the number of molecules to transfer an electron or a hydrogen atom to the nitrogen-centred DPPH radical. The peroxyl radical is performed with a fluorescent probe for the oxygen radical absorbance capacity (ORAC) procedure to create a nonfluorescent product. That product is quantitated by fluorescence. The FRAP, TEAC, and CUPRAC procedures represent electron transfer-based tests, which modify colour when bis (neocuprine) Cu_2_+Cl_2_, Fe_3_^+^ (2,4,6–tripyridyl–s–triazine) 2Cl_3_, and 2,2–azinobis (3–ethylbenzothiazoline–6-sulfonic acid) radical cation (ABTS+) probes are diminished. The antioxidant capacity of the leaves and stems of bilberry was highlighted as a result of their capability to lower the DPPH radical and transition metal ions in the Folin–Ciocalteu assay, and a defensive effect against polyunsaturated dietary lipids in the first steps of gastric digestion. Bilberry leaves present an antioxidant potential (79.30 and 59.58 mM TE/100 g DM in ABTS and FRAP tests) more substantial than that of fruits (35.34 and 26.81 mM TE/100 g DM in ABTS and FRAP assays). On the fruits and leaves of *V. myrtillus*, water, ethyl, and ethanol extraction were performed to test their antioxidant activity. The results showed that the water extract from leaves possesses high antioxidant activity, which is in interrelationship with the large concentration of total phenols [32].

### 2.4. Antimicrobial Activity

It is acknowledged that most phenolic substances possess antibacterial properties. The most relevant are flavonoids, phenolic acids, tannins, and lignans [53]. Regarding bilberry, it has been proven that the antimicrobial activity is determined by the flavonoid fraction, especially anthocyanins. The bilberry extract (BE), along with other purified polyphenolic extracts of berries, presents antimicrobial effects against human pathogenic bacteria, including *Salmonella* and *Staphylococcus aureus* [54]. Bilberry phenolic extracts from fruits and leaves possess antimicrobial effects, principally in avoiding urinary tract infections (UTI) [32,55].

Moreover, BE demonstrated competence in inhibiting *Helicobacter pylori* expansion and some distinct Gram-positive bacteria: *Bacillus, Clostridium*, and *Staphylococcus* strains. Conversely, there is evidence that pure phenolic compounds can restrict and inhibit only Gram-negative bacteria such as *Salmonella* species and *Escherichia coli*. Furthermore, the hydroxylation activity of the phenolic compounds is linked to the Gram-negative bacteria enumerated before [45]. Hence, it can be expected that phytocomplexes or whole fruits perform more efficiently as antimicrobials, with all respect for the purified compounds [36].

Toivanen et al. established that juices obtained from *V. myrtillus* exhibit potential contra pneumococcal infections activated by *Neisseria meningitides* with a 63% increase in inhibition at a concentration of 10 mg/mL [56]. Huttunen et al. focused on the inhibitory action of wild berry juice fragments constructed mainly of sugars and small amounts of small-size phenolics on *Streptococcus pneumoniae* linked to human bronchial cells. They concluded that the highest concentration in their antimicrobial assay (over 85 mg/g) was exceptionally effective, and the development of *Streptococcus pneumoniae* was inhibited by the extract of *V. myrtillus* [44,57]. Another study on the antimicrobial effects of the lipid extracts of two berries, bilberry and lingonberry, by Klavins et al. showed that the fractionated lipids of both berries possess antimicrobial activity. Bilberry and lingonberry lipids were extracted from their seeds using CHCl_3_. The texture and presentation of the extracted lingonberry lipids were a semisolid yellow-coloured paste, while the lipids extracted for bilberry presented a wet texture and a dark green colour [58]. The obtained extraction return was 2.84 g of lipids/100g for bilberry and 2.22 g/100g for lingonberry. Larger seeds than lingonberry characterize bilberries, which represent the fruit’s energy storage, and fatty acid contents are higher; up to over 60% of the lipids. Because of the size of the seeds, extraction yield in the case of bilberries was higher than lingonberries. Lingonberry incorporates elevated amounts of alkanes and alkenes (17%), isoprenoids (17%), and phenolic acids (31%). Meanwhile, bilberry contains higher amounts of di–ketones (3%), which are not found in lingonberries [58].

The antimicrobial activity of total berry lipid extracts and extract fractions was tested against six human or opportunistic pathogens using the agar-well diffusion method. Microorganisms used in this assay were: *S. aureus* MSCL 334, *Streptococcus pyogenes* MSCL 620, *S. epidermitis* MSCL 333, *E. coli* MSCL 332, *Pseudomonas aeruginosa* MSCL 331, and *Proteus mirabilis* MSCL 590. The antibiotic used for the positive control was gentamicin (15 mg/mL). All the samples subjected to tests were at a concentration of 15 mg/mL. The results showed that the obtained fractions from bilberry and lingonberry presented antimicrobial activity against *S. aureus*, *S. pyogenes*, *S. epidermidis*, and *E. coli*, but showed no effect against *P. mirabilis* and *P. aeruginosa*. The inhibition zones reached about 20 mm for bilberry chloroform fraction with *S. aureus*. The antibiotic used for the positive control, gentamicin, exhibited powerful antimicrobial effects on all tested bacterial strains. The solvent used for the negative control was DMSO (Dimethylsulfoxide) [58].

BE and its fractions, principally rich in β–sitosterol, containing 280 mg/g and 70 mg/g, exhibited high inhibition potential against *S. aureus* strains. Therefore, chloroform and ethyl acetate fractions had 20 and 15 mm inhibition zones. The chloroform and ethyl fractions were also characterized by an inhibitory effect against other Gram-positive bacteria, including *S. pyogenes and S. epidermidis*. All tested bilberry fractions showed an inhibitory effect against *S. pyogenes*, except the most non-polar hexane fraction. The tested berry lipid fractions had an inhibitory effect against only one Gram-negative bacteria, *E. coli*. Inhibition zones of 11 mm were presented by the bilberry ethyl acetate/methanol and methanol fraction [58].

### 2.5. Health Benefits

*V. myrtillus* fruits have been consumed since ancient times and have become popular nowadays, representing an essential part of the everyday diet and used in diverse popular medicines [59]. These berries are well known as plentiful natural sources of polyphenols and more bioactive compounds with health effects. The consumption of bilberry fruits presents a broad spectrum of health benefits, such as antioxidant, anti-proliferative, anti-inflammatory, anti-obesity, cardioprotective, hypoglycemic, ocular, and antibacterial effects, as presented in Table 2 [44,60,61,62].

#### 2.5.1. Anti-Inflammatory Effect

The anti-inflammatory activities of bilberry fruits and their extracts are based on their high anthocyanin content. Mirtoselect is a commercial BE standardized to a concentration of anthocyanins of 40%. This extract showed a complex anti-inflammatory response in lipopolysaccharide-activated macrophages. Its activity consists of attenuating pro-inflammatory cytokines (including TNF–α, IL–1β, IL–6, and COX–2) and enfeeblement of multiple lipopolysaccharide-induced chemokines and IL receptors almost to control levels [75].

The feedback of the human THP–1 monocytic cell after the treatment with anthocyanin-rich bilberry extract to inflammatory stimulation was more diverse. BE attenuated the IFN–γ–induced signal protein activation, cytokine secretion, and pro-inflammatory gene expression, since it increased TNF–α–induced reactions [67]. This effect offers a distinct role of anthocyanins in inflammatory modulation that can diversify from target to target. In vitro results are presented in various clinical studies based on inflammatory diseases; for example, after ingesting 500 g of bilberries daily, patients diagnosed with gingivitis presented reduced inflammatory cytokine levels and minor bleeding on probing, representing a routine dental clinical parameter of inflammation [76]. In another study performed on patients suffering from increased cardiovascular risk, after ingesting 330 mL of bilberry juice daily for four weeks, some of the participants’ inflammatory blood markers substantially decreased in comparison with the placebo group (C–reactive protein, IL–6, IL–15, and monokine induced by INFγ), although others remained unaffected. Surprisingly, an expansion of TNF–α was also observed [77].

Subjects with metabolic syndrome were tested after eating 400 g of fresh bilberry for eight weeks. Reductions of several inflammatory parameters (C-reactive protein, IL–6, IL–12, and LPS) were registered and typically showed lower inflammation scores [76]. In metabolic syndrome subjects, complementary anti-inflammatory evidence was accomplished with a supplement consisting of purified bilberry and blackcurrant anthocyanins Accordingly, it is a fair assumption that these components are the predominant active principle behind the sympathetic effects of *V. myrtillus* on cardiometabolic risk factors [77].

Inflammatory bowel disease, a functional gastrointestinal disorder [78], often lacks proper treatment, having several adverse effects. The necessity to find a suitable and effective treatment is reasonable. Evidence shows that a dysregulated immune response referring to intestinal microbiota causes the inflammation. Experiments in vitro showed that bilberry fruit extract and the isolated anthocyanins could constrict the inflammatory response of human colon epithelial cell cultures aroused by IFN–γ/IL–1β/TNF–α [77]. An open, non-controlled, and non-blinded trial performed on patients with moderate ulcerative colitis revealed that over 90% of the subjects presented positive feedback to a blend made of dried bilberry powder and concentrated bilberry juice, with a number over 60% of the patients that achieved remission [38,70].

Several diseases are influenced in their development and progression by inflammation, both acute and chronic. Schink et al. [62], in a selection of 99 ethanolic plant extracts, established that *V. myrtillus* presented intense anti-inflammatory activity linked with high cell viability in THP–1, HeLa–TLR4, and HEKTLR2/HEK–TLR4 cell lines. Anthocyanin-rich extracts from *V. myrtillus* displayed anti-inflammatory effects contra liver inflammation in mice, conducting to the suppression of LPS-induced inducible nitric oxide synthase (iNOS), TNF–α, IL–1β, and IL–6 transcripts, and iNOS, TNF–α, and NF–κB protein levels [63]. Moreover, bilberry also showed its ability to decrease serum C-reactive protein (CRP), IL–6, IL–12, and LPS levels, and lowered genes linked with the TLR pathway in individuals who presented with metabolic syndrome [44,62,76].

#### 2.5.2. Anti-Cancer and Anti-Proliferative Activity

Several studies performed on bilberry extracts to determine their antioxidant activity helped create treatment programs against different types of cancer. Portions containing flavonoids, catechins, and organic acids extracted from bilberry fruits were potent in inhibiting the development of cervix epitheloid carcinoma, breast carcinoma, and colon adenocarcinoma cell lines (with IC50 extending between 125 to 300 mg/mL) [79]. The growth of three colon cancer cell lines (Caco–2, HT–29, and HCT–116) was restrained by the whole bilberry extract [22]. Additionally, the progression of prostate cancer in cell lines was obstructed with whole bilberry extract and the standardized mixture of its principal phenolic compounds [73].

The powder obtained from the whole bilberry suppressed the proliferation, viability, invasion, and migration of oral squamous carcinoma cell lines in the zebrafish model [80]. The fruit extract also prohibited early preneoplastic liver cell lesions in rats [38,81].

#### 2.5.3. Antidiabetic Effect

Studies performed on human individuals showed that fasting serum hippuric acid is enhanced after ingesting anthocyanin-rich bilberries. This can assist in the beneficial effect of bilberry consumption through its ability to improve cell function and glycaemic control in diabetic patients [60]. Angiogenesis represents the physiological progression through which the circulatory system is improved by developing new blood vessels from pre-existing ones made in the earlier stage of vasculogenesis. The improvement of the vasculature is protracted by angiogenesis with processes of sprouting and splitting [82]. Studies demonstrated that irregular angiogenesis is the cause of multiple ocular diseases, such as neovascular glaucoma, diabetic retinopathy, and age-related macular degeneration [83]. In accordance, bilberries exhibit an inhibitory effect on angiogenesis and can involve indirect antidiabetic properties [83,84].

Bilberries possess the ability to take an angiotensin-converting enzyme role in human angothelial cells and demonstrated their effectiveness on cardiovascular disease together with their antidiabetic properties [74]. Studies revealed that bilberry’s inhibitory property on angiogenesis results from the effects of their principal components, including malvidin, delphinidin, and cyaniding [84].

*V. myrtillus* leaf has a notable role in reducing hyperglycaemia in diabetic animals [85]. They can delay or even inhibit the initial stage of early diabetic retinopathy. Furthermore, bilberries enhanced diabetes-induced endothelial permeability in addition to their linked loss of tight junction integrity and vascular endothelial growth factor (VEGF) upregulation [72]. α–glucosidase represents one of the most important enzymes linked with diabetic disease, and studies reported that bilberries could inhibit α–glucosidase enzyme [65]. Bilberries can be considered a treatment strategy because they significantly reduce oxidative stress in rats [86]. A study showed that the consumption of bilberry could improve insulin secretion and decrease blood glucose [31].

Studies performed on animal models to show bilberry’s antidiabetic activity demonstrated that blood glucose was meaningfully lowered after a treatment session with a phenolic-rich extract and an anthocyanin-enriched part from bilberry extract in mice [87]. One study by Asgary et al. [88] demonstrated the antihyperlipidemic and anti-hyperglycaemic effects of bilberry fruits in alloxan-induced diabetic rats. Lowering the branched-chain amino acids and upgraded lipid formation was stated as the principal antidiabetic mechanism of bilberries [89], alongside the capability of its leaves to treat dyslipidaemiae linked with impaired TG-rich lipoprotein clearance in rats [86]. *V. myrtillus* fruits lowered markers of diabetic retinopathy, such as (VEGF) expression and degradation of zonula occludens–1 and claudin–5 in diabetic rats. In conclusion, bilberry extract has the ability to prevent or delay the beginning of early diabetic retinopathy [64,72].

#### 2.5.4. Anti-Obesity Effect

The anti-obesity techniques of bilberries may comprise a decline in lipid absorption, a diminish in proliferation and differentiation of preadipocytes, a cutback in lipogenesis, an escalation in lipolysis, and restriction of pro-inflammatory adipokine secretion [90,91]. The ability of *V. myrtillus* to decrease lipid aggregation with an adjuvant down-regulation of peroxisome proliferator-activated receptor gamma (PPARγ) was presented by Kowalska et al. [90]. Together with PPARγ, other elements were found in mouse embryo 3T3–L1 adipocytes, including: enhancer-binding protein alpha (C/EBPα) and sterol regulatory element binding transcription factor 1 (SREBP1c). Another ability of bilberries found by the researchers was the potential to restrain the expression of adipocyte fatty acid-linking protein (aP2) and resistin [44,92].

#### 2.5.5. Cardioprotective Activity

Cardiovascular diseases (CVD) represent one of the most dominant causes of death and are a fundamental focus of research and treatment [44,93]. Atherosclerosis is a chronic inflammatory disorder linked with oxidative processes. It represents the principal cause of CVD, along with myocardial infarction (MI), stroke, heart failure, and claudication [94]. An investigation performed on an apolipoprotein E–deficient (apo E–/–) mice model of atherosclerosis showed that a dietary supplement composed of bilberry anthocyanin-rich extract, including over 50% of pure anthocyanins ingested for two weeks, was able to lower plasmatic total cholesterol (about 20%) and hepatic triglyceride levels (about 30% in the liver). However, the plasma antioxidant capacity remained unaltered [95]. Moreover, another study on the same subject showed that these bilberry extracts act on the inflection of gene expression implicated in angiogenesis in the aortas of apo E–/– mice [96]. The potential favourable consequence of *V. myrtillus* has also been studied on the evolution of obesity in mice subject to a high-fat diet (HFD) [71]. The animals were fed with 5% or 10% (*w*/*w*) of whole bilberries in HFD for about three months. They presented a reduced glucose level and lower blood pressure related to mice fed only with HFD. Furthermore, adding bilberries to HFD showed a reduced level of various inflammation parameters, but the insulin status in mice was not affected by adding bilberries to their diet.

Studies performed on humans showed that bilberry leaves possess a cardioprotective activity [97]. Erlund et al. [98] conducted human research to examine the effects of consuming a mixture composed of 100 g of bilberries and nectar containing 50 g of crushed lingonberries for two months, and the mix was consumed daily. The study was performed on well-established risk factors of CVD: platelet function, HDL cholesterol, and blood pressure. Furthermore, the subjects ingested strawberry puree or blackcurrant and cold-pressed chokeberry and raspberry juice on alternating days. No significant changes were observed in plasma biomarkers of platelet activation, fibrinolysis, or coagulation. Meaningful changes were registered in the systolic blood pressure, which was lowered, and the serum HDL cholesterol concentrations were raised. A recent study presented critical changes in lipid metabolites caused by an intake of bilberry in combination with fish and wholegrain in subjects suffering from coronary diseases [32,99].

## 3. Cranberry (*V.* sect. *Oxycoccus*)

### 3.1. History, Nomenclature and Production

Cranberry is a diploid fruit [100], a woody perennial that produces vertical stems [101], belonging to the *Ericaceae* family, genus *Vaccinium*. *V.* sect. *Oxycoccus,* known as cranberry. It is divided into four species: *V. erythrocarpum* (southern mountain cranberry*)*; *V. macrocarpon* (large cranberry, American cranberry, or bearberry)*; V. microcarpum* (small cranberry)*;* and *V. oxycoccos* (common cranberry or northern cranberry) [102]. The world’s most significant cranberry production is in North America, with an annual output of 436.691 tons, thus representing 58% of the world’s total cranberry production [103]. Acid soils and the peat of bogs, swamps, wet shores, and occasionally poorly drained upland meadows, are the perfect environments for growing fruits from the subgenus *V. Oxycocc*us, particularly these two species: *V. macrocarpon* and *V. oxycoccos* [104].

Cranberry (*V. sect. Oxycoccus*) is one of the most cultivated fruits in the U.S., with a global production of over 663.000 tons in 2020; this had a 108% increase from 2000 to 2020 (FAO). Based on the production and consumption of cranberry, this will also present a constant growth trend, with an increase of 44.4% (958107 tons) by 2030. In 2007, around USD 75 M was obtained from agricultural receipts [105]. In addition, considering the last 20 years of cranberry production and using Excel 2022 and FORECAST function, we predicted production until 2030 based on existing values that follow a seasonal trend. A confidence interval was estimated using interval values from 2000–2020. The interval was defined by its lower and upper bounds from 2020 until 2030. Seasonality was considered to be one decade. The confidence interval was expressed as a percentage (95%). The percentage reflects the confidence level. According to data analysis from FAO data, cranberries forecast may reach a value of 0.93 ± 0.14 M tons by 2030. Figure 2 shows a constant forecast and trendline that will increase in the following years.

The American cranberry (*V. macrocarpon*) and the European cranberry (*V. oxycoccos)* are the most common. The American fruit is two times larger than the European cranberry, which is between 0.6 and 1.2 cm. Cranberry juice drinks, sauce, sweetened, dried cranberries, and pemmican cranberry (a high-protein combination of crushed cranberries, dried deer meat, and melted fat) are some of the products obtained from cranberry.

### 3.2. Blueberries Constituents

The most important bioactive compounds in cranberries are anthocyanins, flavonols, flavan–3–ols (catechins), PACs, benzoic and phenolic acids, nonflavonoid polyphenols (phloridzin), terpenes, sterols [106], and several essential micro- (Fe, Mn, Zn, Cu, Mo, B) and macro-elements (N, P, K, Ca, Mg, S), as evidenced in Table 3 [107].

*V. macrocarpon* is an essential healing agent and food because of its composition, flavonoids being its major compounds among over 150 compounds, including 13 anthocyanins, 16 flavonols, and 26 phenolic acids and benzoates [108].

#### 3.2.1. Phenolic Acids

Cranberries’ therapeutic potential, in the case of both species *V. macrocarpon* and *V.oxycoccos*, comes from their antioxidant effect, which is due to its many phytochemicals. Thus, a cranberry mixture or solution contains several classes of phenolics after the extraction process. The method used for quantification and other characterisation includes High-performance Liquid Chromatography (HPLC), Ultraviolet-Visible (UV/Vis), and Gas Chromatography (GC) [108].

In cranberry, high quantities of benzoic acid and low quantities of 2,4–dihydroxybenzoic acid, *p*–h–hydroxybenzoic, and *o*–hydroxybenzoic acids (salicylic acid) can be found. Pınar Kalın et al. found some of the main phenolic compounds in *V. macrocarpon* extract using the Folin–Ciocalteu method. Based on a standard gallic acid curve, they determined the quantity of total phenolics and the total phenolic content in lyophilised aqueous cranberry extract. Some of the main phenolic compounds identified were p–coumaric acid (13.0 mg/kg), *p*–hydroxybenzoic acid (55 mg/kg), kaempherol–3–*O*–glucoside (11.0 mg/kg), caffeic acid (5.0 mg/kg), and ellagic acid (3.0 mg/kg) [110,111]. The quantity of phenolic compounds depends on the cultivar and berry ripening; therefore, total phenolic acids rise to form the lowest content of 327 mg/100 g dry matter (dm) in ”Pilgrim” to the highest content of 649 mg/100 g dm in “Howes” [111]. The most present compounds in all cranberry cultivars (Pilgrim, Stevens, and Ben Lear) were caffeoyl hexoside (representing 34.6% to 54.1% of total phenolic acids) and caffeoyl dihexoside (from 16.5% to 35.0%) [112].

#### 3.2.2. Anthocyanins and Proanthocyanidins

Anthocyanins belong to the polyphenol class and are found in most black fruits (blueberry, black currants, cranberry) and vegetables (red cabbage, radish). Studies in vivo and on experimental animals showed that anthocyanins decrease insulin resistance, protect pancreatic cells against necrosis, and increase glucose uptake by tissues in streptozotocin-induced diabetic rats and mice. Additionally, they improve insulin secretion (which can help people with type two diabetes), and particularly help women with UTIs, which are one of the most common bacterial infections [113]. Studies showed that PACs found in cranberries inhibit the adherence of p-fimbriated *E. coli* on the mucosal surface of the urinary tract, obstructing bacterial reproduction [114].

Anthocyanins (Table 4) give cranberries their red colour, and they are natural water-soluble pigments found in amounts 6 to 10 times higher in the peel in contrast to the pulp of the small berry (*V. oxycoccos*) [111]. Climate, cultivar, growing locations, and genetic traits are some of the things that influence the content of anthocyanins in cranberry. The anthocyanins level significantly increases during the maturation process in cranberry. For example, in the fruit cultivars “Pilgrim”, “Stevens”, and “Ben Lear”, anthocyanins increase by 57.3%, 47.0%, and 30.0%, respectively, from the immature stage to the mature stage [112].

A differentiating compound in cranberry is the A-type PAC, which hinders the in vitro bond of *E. coli* to uroepithelial cells [111,114]. Jacques Masquelier discovered PACs in the 1940s and gave them the name ‘’vitamin P’’. An essential characteristic of PACs is their capacity to precipitate proteins and polypeptides, acting more on those with a high proline level. A-type polymers are less frequent and have at least one intermolecular bond between O7 and C2, in addition to the C–C bond (Figure 3) [102]. One of the most important constituents of PACs is epicatechin (−), whereas epicatechin, catechin (+), and (epi) gallocatechin have an important role but are present only in trace amounts [115]. Compared to *V macrocarpon,* which has a range of 2.80–5.05 mg/100 g fw (fresh weight), *V oxycoccus* has a range of 0.55–1.94 mg/100 g fw [111,116].

#### 3.2.3. Flavonoids

Anthocyanins, flavonols, and flavan–3–ols are the elementary classes of flavonoids found in cranberry. Because of their polyphenolic structure, they play a vital role in antibacterial, antiviral, anticarcinogenic, anti-inflammatory, and vasodilatory activities, as well as in defence mechanisms and antioxidant activity. Cranberry fruit has a significant concentration of flavonoids, such as kaempferol, myricetin, and quercetin in the peel; the ripening stage determines this concentration [102,108,111].

**Table 4 molecules-28-01533-t004:** Phytonutrients in cranberry (adapted from [111,112,117,118]).

Phytonutrient	Name	Content
**Anthocyanins**	Delfinidyn derivatives *	31.27–43.87 mg/100 g dm
	Delfinidyn-3-O-glucoside *	1.1–1.8 mg/100 g dm
	Cyanidin derivatives *	442–967 mg/100 g dm
	Cyanidin-3-O-galactoside *	119.9–180.0 mg/100 g dm
	Cyanidin-3-O-glucoside *	5.5–7.3 mg/100 g dm
	Cyanidin-3-O-arabinoside *	64.5–95.6 mg/100 g dm
	Peonidin-3-O-galactoside *	131.3–310.3 mg/100 g dm
	Peonidin-3-O-arabinoside *	42.9–95.2 mg/100 g dm
	Peonidin derivatives *	192–666 mg/100 g dm
	Malvidin derivatives *	29.85–58.85 mg/100 g dm
	Malvidin-3-O-arabinoside *	1.4–1.9 mg/100 g dm
	**Total anthocyanins ***	**695–1716 mg/100 g dm**
**Phenolic acid**	p–Coumaric acid ***	2–245 µg/g dw
	p-Coumaroyl hexose *	8.6–13.9 mg/100 g dm
	p-Coumaroyl hexose isomer *	3.6–50.0 mg/100 g dm
	p-Coumaroyl derivatives *	210–451 mg/100 g dm
	Chlorogenic acid *	72.00–129.62 mg/100 g dm
	Caffeic acid ***	5–123 µg/g dw
	Caffeoyl hexoside *	92.7–190.2 mg/100 g dm
	Caffeoyl hexoside isomer *	10.9–17.5 mg/100 g dm
	Caffeoyl and derivatives *	39.93–68.28 mg/100 g dm
	Ferulic acid ***	4–39 µg/g dw
	**Total phenolic acid ***	**327–649 mg/100 g dm**
**Flavonols**	Myricetin-3-O-galactoside *	156.5–348.4 mg/100 g dm
	Myricetin-3-O-glucoside *	1.8–6.6 mg/100 g dm
	Myricetin-3-O-pentoside *	6.3–55.6 mg/100 g dm
	Myricetin-3-O-glucoronide *	19.0–38.5 mg/100 g dm
	Myricetin-arabinoside ***	8–273 µg/g dw
	Sinapoyl derivatives *	4.36–5.82 mg/100 g dm
	Myricetin derivatives *	496–926 mg/100 g dm
	Quercetin-3-O-galactoside *	294.6–375.8 mg/100 g dm
	Quercetin-3-O-pentoside *	21.2–122.9 mg/100 g dm
	Quercetin-3-O-glucoside *	4.8–11.5 mg/100 g
	Quercetin-p conmaroylhexoside *	1.3–13.3 mg/100 g dm
	Quercetin-3-O-rhamnoside *	6.2–13.3 mg/100 g dm
	Quercetin-rutinoside *****	12.0 mg/100 g fw
	Quercetin-acetyl-glucosidase *****	13.58 mg/100 g fw
	Quercetin derivatives *	107–225 mg/100 g dm
	Methoxyquercetin hexoside *	1.7–25.7 mg/100 g dm
	Methoxyquercetin pentoside *	3.4–61.0 mg/100 g dm
	Methoxyquercetin derivatives *	33.31–43.04 mg/100 g dm
	**Total flavonols ***	**643–1088 mg/100 g dm**
**Flavan-3-ols and proanthocyanidins**	(+)-Catechin *	2.79–7.53 mg/100 g dm
	(−)-Epicatechin *	27.46–56.84 mg/100 g dm
	A-type PA-dimer *	16.94–32.07 mg/100 g dm
	A-type PA-trimer *	27.82–76.94 mg/100 g dm
	A-type PA-tetramer *	41.51–65.61 mg/100 g dm
	B-type PA–dimer *	12.62–36.75 mg/100 g dm
	B-type PA–trimer ******	0.04–2.93 mg/100 g fw
	Polymeric proanthocyanidins *	651–1109 mg/100 g dm
	Sinapyl hexose *	2.0–3.3 mg/100 g dm
	**Total flavan–3–ols and proanthocyanidins ***	**860–1283 mg/100 g dm**
**Triterpenoids**	Ursolic acid *	1044–1714 mg/kg dm
	Oleanolic acid *	894–1137 mg/100 g dm
	Betulinic acid *	635–824 mg/kg dm
	**Sum Triterpenoids ***	2892–3671 mg/kg dm
	**Total Sterols (β–sitosterol and stigmasterol) ******	**107.83 mg/g fw**

dm—dry matter; dw—dry weight; fm—fresh matter; fw—fresh weight; methods used to identify phytonutrients: * LC/MS Q-TOF—Liquid Chromatography/Mass Spectrometry Quadrupole Time-of-Flight; *** HPLC/ESI-MS/MS—high-performance liquid chromatography/electrospray ionization tandem mass spectrometry; **** HPLC-DAD—high-performance liquid chromatography-diode array detector; ***** HPLC-PDA—high-performance liquid chromatography/photodiode array; ****** UHPLC UV/MS—ultra-high performance ultra-violet mass spectrometer.

During this ripening, flavonol levels increased by 25%, 9%, and 1% in ’’Pilgrim’’, ’’Stevens’’, and ‘’Ben Lear’’, respectively [102,108,111]. Flavonoid content in three cranberry cultivars (Stevens, Pilgrim, Ben Lear) showed small differences. Stevens had the highest amount (142.1 mg/100 g fw), which was similar to Pilgrim (138.6 mg/100 g fw), and the lowest amount of flavonoids was found in the Ben Lear cranberry cultivar (114.2 mg/100 g fw) [119]. Fresh cranberry juice has a medium of 400 mg of total flavonoids (56%) and other phenolic compounds (44%) per litre [108].

#### 3.2.4. Triterpenoids

Cranberry fruit is also rich in triterpenoids; they have a complex cyclic structure and exist as carboxylic acids, alcohols, or aldehydes [111,120]. The most important compounds in this class are ursolic acid, which has an anti-inflammatory effect and can inhibit liver and breast cancer. Ursolic acid’s isomers, especially cis–3–*O*–*p*–hydroxycinnamoyl ursolic acid, trans–3–*O*–*p*–hydroxycinnamoyl ursolic acid, and oleanolic acid, which are found in the wax on the peel, present anti-inflammatory, antitumor and anticancer effects. *V. oxycoccus* contains ursolic acid, which defends against oxidative damage and lipid oxidation [111,121]. Wu et al. (2020) identified in cranberry extract significant quantities of ursolic acid (372.97 mg/g), oleanolic acid (79.16 mg/g), and β–sitosterol and stigmasterol (107.83 mg/g) [118]. The cultivar ‘’Franklin’’ showed the highest amount of triterpenoids (3671 mg/kg dm), and the lowest content was in the “Pilgrim’’ cultivar (2892 mg/kg dm) [111].

Several studies have shown that ursolic acid can inhibit tumour colony formation in HT–29 and HCT116 human colon tumour cell lines. The growth of tumour cell lines, including HT–29 colonic cells, is inhibited by triterpenoid esters (cis– and trans–3–*O*–*p*–hydroxycinnamoyl ursolic acid) isolated from cranberry juice [118]. Another study showed that the peel samples of *V. macrocarpon* and *V. oxycoccos* contain the highest amount of the triterpenes and phytosterol compounds (11,109.86 ± 166.27 µg/g and 9582.45 ± 143.74 µg/g). Higher levels of triterpenoids and phytosterols were found in the cultivar “Stevens” (*V. macrocarpon*) and lower amounts in small cranberry (*V. oxycoccos*). The ursolic acid found in fruit samples was 64.80% for the large cranberries and 62.64% for the small cranberries [121].

### 3.3. Antimicrobial Effect

Four American cranberry (*V. macrocarpon*) cultivars, ‘’Stevens’’, ‘’Pilgrim’’, ‘’Ben Lear’’, and ‘‘Black Veil’’, were grown in the same experimental field of Kaunas Botanical Garden of Vytautas Magnus University and analysed.

The antimicrobial activity was performed on Gram-positive *Listeria monocytogenes* (ATCC 19117), *Bacillus cereus* (ATCC 10876), *B. subtilis* (ATCC 6633), *Micrococcus luteus* (ATCC 9341), *Enterococcus faecalis* (ATCC 29212), and *S. aureus* (ATCC 25923), and Gram-negative *E. coli* (ATCC25922), *Enterobacter aerogenes* (ATCC 13048), *Salmonella typhimurium* (ATCC 14028), and *Slm. agona* bacteria. Clavulanic acid was used at 30/10 μg/mL for positive control and acidified ethanol for negative control. The agar well diffusion method was used to test the antimicrobial effect. All extracts exposed antimicrobial properties, and the most effective was Pilgrim. This method showed that *B. cereus* and *M. luteus* were the most sensitive, with an inhibition zone between 2.28 and 2.24 cm. All test cultures were sensitive to the clavulanic acid 30/10 μg sensi-disc, omitting *B. cereus*, which was more responsive to berry extract. Additionally, eight strains of yeast isolated from dairy products, industrial air, and equipment washing water were used: *Debaryomyces hansenii*, *Trichosporon cutaneum*, *Kluyveromyces marxianus* var. *lactis*, *Saccharomyces cerevisiae*, *Candida parapsilosis*, *Torulaspora delbrueckii*, *Pichia kluyveri*, and *Rhodotorula rubra*. Cranberry extracts did not affect the multiplication of any of the eight yeast species [122,123].

In addition, cranberries have an essential effect on the inhibition of oral pathogenic bacteria, such as *Streptococcus mutans*, *Streptococcus gordonii, Streptococcus sobrinus, Actinomyces naeslundii, Fusobacterium nucleatum, Aggregatibacter actinomycetemcomitans, Porphyromonas gingivalis,* and *Enterococcus faecalis*. The method used to prove the antimicrobial effect on oral pathogenic bacteria was the agar diffusion assay. In addition, anthocyane and proanthocyanidin found and extracted from cranberries were antibacterial, but *S. mutants* was not supported by cranberry extract, inhibiting *E. faecalis* partially. For positive control, chlorhexidine (CHX) 2% was used, and this study revealed that antibacterial efficiency is influenced by incubation time and concentration. Cranberry juice at 90% concentration and exposure times of 60 s could suppress four out of eight pathogenic species (*S. sobrinus, F. nucleatum, A. actinomycetemcomitans,* and *P. gingivalis*) [124].

### 3.4. Medicinal Effect of Cranberry Consumption

Cranberries have many medicinal applications, and their consumption can prevent many diseases (Table 5), such as urinary tract inflammation, cystitis, oxidative stress, cardiovascular diseases, obesity, type 2 diabetes, microbial infections, tooth decay, periodontitis, and cancers [111].

Gastric cancer: is one of the most frequent types of cancer for all sexes and ages. According to the World Health Organization (WHO) [125], *H. pylori* are often linked with the evolution of most gastric ulcers. However, eradicating antibiotics is not a strategy because *H. pylori* colonisation can show some health benefits in reliable situations. Cranberry has many bioactive compounds, including PACs with A–type double linkages. In addition, the presents of the PACs that have an anti-adhesion effect on bacteria are important because they inhibit the initial stage of the infection instead of killing the bacteria [125]. To demonstrate the anti-adhesion effect, Burger et al. (2000) conducted a study on the adhesion of three strains of *H. pylori* (BZMC-25, EHL-65, and 17874) on gastric mucus acquired from a stomach taken after post-mortem surgeries. Non-dialysable material (NDM) was a cranberry juice dialysed against distilled water in dialysis bags; after six days of dialysis, NDM was lyophilised. NDM at 100 μg mL^−1^ concentrations inhibited *H. pylori* BZMC-25 and constrained adhesion to human gastric mucus. The inhibitory effect was dose-dependent, and 50% of the inhibitory concentration relied on BZMC-25, EHL-65, and 17,874 strains [126].Obesity: many studies show that cranberry reduces lipid accretion by lowering the mRNA level of some genes associated with fatty-acid-binding protein, lipoprotein lipase, fatty acid synthase, hormone-sensitive lipase, and perilipin 1. However, one of the most important targets for obesity prevention is to decrease leptin and increase adiponectin gene expression and adipocytokines secretion A [127]. In vivo studies on obese diabetic mice have shown that a dose of 5% and 10% of cranberry powder in the diet for six weeks and a cranberry extract (200 mg/kg) for eight weeks increased HDL-cholesterol level and decreased insulin and glucose level. The decreased hepatic, intestinal, and plasma triglyceride accumulation reduced oxidative stress and inflammation [128]. In a study on mice, Kunkel et al. (2012) showed that ursolic acid identified in cranberry reduces obesity, improves glucose tolerance, and decreases hepatic steatosis [129].Diabetes: cranberries are rich in phenolic compounds such as quercetin which can inhibit gastric assimilation of glucose in the porcine model [130]. Besides quercetin, they also contain myricetin, which may inhibit glucose transporter type 4 mediated glucose assimilation by rat adipocytes [140]. Patients with type 2 diabetes were involved in two 12-week studies to prove that cranberry bioactive constituents help features of metabolic syndrome and diabetes. Cranberry juice significantly reduces restrained glucose, while cranberry extracts mark down total and LDL–cholesterol. Some studies show that dried low-calorie cranberry-enriched high-fat meal challenge induced postprandial glycaemia, inflammation, and lipid peroxidation in diabetes [131]. Another study shows that daily consumption for 12 weeks of cranberry juice (240 mL) and blueberry extract (9.1–9.8 mg of anthocyanins) for 8 to 12 weeks improved glucose control in type 2 diabetes subjects [141].Urinary tract infection (UTI): is a bacterial infection which affects young and sexually active women. Approximately 1 in 3 women will experience an episode of UTI, requiring antibiotics [114]. In vitro studies show that A-type PACs constrain the adhesion of P-fimbriated uropathogenic *E. coli* to uroepithelial cells [116]. Antibiotics to treat UTIs have been claimed to be very functional, but these antibiotics can damage intestinal microbiota and cause resistance among uropathogens. PACs are usually catabolised by the colon microbiota to give bioactive phenolic metabolites. This phenolic metabolite may be the responsible compound behind the anti-adhesion effect. D–mannose also inhibits adhesion on uroepithelial cells in vitro. Another helpful characteristic of cranberry juice is the pH of 2.5, which causes changes in urine’s physical properties (acidification) [132].Periodontitis: is an inflammatory disease which affects tooth-supporting tissues (periodontal ligament and alveolar bone) and is induced by Gram-negative anaerobic bacteria. NDM can inhibit the proliferation of *P.gingivalis*, *T.forsythia*, and *T.denticola* in periodontal pockets. Unusual production of cytokines by host cells caused by periodontopathogens damage tooth-supporting tissues; studies have demonstrated that cranberry was an excellent inhibitor of pro-inflammatory cytokine and chemokine replies caused by lipopolysaccharides [124,133,134].

## 4. Another Vaccinium sect. *V. membranaceum*

### 4.1. History, Nomenclature, and Production

Approximately 400 species in the diverse *Vaccinium* genus are dispersed quasi-globally, with Antarctica and Australia as the exceptions. The genus is commonly described as shrubs or perennial vines with fleshy and moderate-sized, usually edible fruit [142,143]. Berries are considered sources of many nutrients, such as minerals, vitamins, sugars, fibre, and other compounds, with health benefits that may vary with the stress plants face [144]. Based on the taxonomical interpretation, up to 14 *Vaccinium* species are described in the Pacific Northwest; some examples are defined in Table 6. These species bear fruits which are often referred to as huckleberries; the same can be said about the more Eastern species, yet they are very much distinct from “true huckleberries”, which are found in the *Gaylussacia genus* [145].

Huckleberries are ericaceous (Ericaceae or Heath family) tetraploid (2n = 48) shrubs with fleshy, flavorfanthocyanicyan-rich, edible berries [146]. Western huckleberries, known as bilberries, mountain huckleberries, gooseberries, big huckleberries, black huckleberries, whortleberries, and thin-leaf huckleberries, are found in the genus Vaccinium section *Myrtillus* and are related to sect. Cyanococcus highbush (*V. corymbosum* L.) and lowbush blueberries (*V. angustifolium* Ait.). This section will focus on huckleberry species, primarily addressing the *V. membranaceum* plant.

The human usage of huckleberries may extend to as far back as 11.000 B.P. (Before Present). They have had significant importance in the cultural and socio-economical life of native peoples. They were eaten raw, cooked, mashed, and dried in the sun as cakes. In the 1800s, berry picking started to grow to an industrial level, taking advantage of the invention of cans, and by the 1920s, commercial companies had been established. In 1932, huckleberries were a substantial crop, with 1 M.T. picked in Montana and marginally less in Washington State. Around this time, an interest in better understanding huckleberries led to Schultz’s chromosome description (2n = 48) in 1944. In the decades that followed, the compositional aspect of these berries became better understood by Kuhlein in 1989, and their list of nutrient values in 2004 by Lee, Finn, and Wrolstad, with research on anthocyanin pigment and phenolic compounds [143,147,148,149].

**Table 6 molecules-28-01533-t006:** Huckleberry species and characteristics, adapted from [149,150,151].

	*V. membranaceum*	*V. parvifolium*	*V. scoparium*	*V. ovalifolium*	*V. ovatum*	*V. caespitosum*	*V. deliciosum*
Names	bilberries, mountain huckleberry, grouseberries, hortleberries, black huckleberry, hortleberries, and thin-leaf huckleberry	red huckleberry, red bilberry	small-leaved huckleberry, grouseberry, dwarf red whortleberry, and red alpine blueberry	oval-leaf blueberry, oval-leaf bilberry, oval-leaf huckleberry, Alaska blueberry	evergreen huckleberry	dwarf blueberry, dwarf bilberry, dwarf huckleberry, and dwarf whortleberry	cascade huckleberry, cascade bilberry or blue huckleberry
Flower	creamy-pink	greenish to pinkish	pink	pink	bright pink	pink	pink
Fruit/Berries	large, shining black, dull black, deep purple, rarely red, 9–13 mm d.	red, occasionally faintly glaucous, 7–9 mm diam, more tart than sweet	translucent red, red, or bluish purple, 4 -6 mm d., soft, tart	bright blue, glaucous, dull purplish black or black, 810 mm d.	purplish-black, 6–9 mm d.	bright blue and glaucous, rarely dull black, 59 mm d., great flavour	blue and glaucous, occasionally dull black, maroon, or red, 9–15 mm d., especially flavorful berries
Distribution	Rocky Mountains from SW-NW Territories S to N California and N Utah	Pacific coast of N America from Alaska to N California, inland to SE British Columbia	SE British Columbia and adjacent Alberta, E to the Black Hills of South Dakota, and south to SW Colorado	Pacific Rim from Central Japan, Kamchatka, Aleutian Islands, S along the Pacific coast to S central Oregon and inland to N Idaho	Pacific Coast from British Columbia to central California	Alaska to Newfoundland, southward along the Atlantic Seaboard to S Maine and S Vermont; in the W, S to the W highlands of Guatemala	Pacific coastal mountain ranges from S British Columbia to N California, Montana and Idaho
Commercial value	food industry: jams, syrups, tea, culinary uses; medicinal uses; cosmetic uses	berries—food industry: jam, jelly, wine; leaves—medicinal use, ornamental use	culinary use	culinary use	the food industry, such as jam; medicinal use; ornamental use	the food industry; ornamental use	food industry
* TP	1.70 mg of GAE/g of FW	0.81 mg of GAE/g of FW	-	2.81 mg of GAE/g of FW	2.84 mg of GAE/g of FW	-	1.41 mg of GAE/g of FW
** TA	1.69 mg of C3G/g of FW	0.11 mg of C3G/g of FW	-	3.07 mg of C3G/g of FW	3.64 mg of C3G/g of FW	-	1.35 mg of C3G/g of FW
*** AC	ORAC 21 µmol of TE/g of FW, FRAP 40.5 µmol of TE/g of FW	ORAC 7.3 µmol of TE/g of FW FRAP 10 µmol of TE/g of FW	-	ORAC 37.8 µmol of TE/g of FW FRAP 76.2 µmol of TE/g of FW	ORAC 41.1 µmol of TE/g of FW—FRAP 70.2 µmol of TE/g of FW	-	ORAC 14.6 µmol of TE/g of FW FRAP 30.2 µmol of TE/g of FW
**** F3 and F	Catechins > 240 µg/g of FW	Catechins > 154 µg/g of FW	-	Catechins > 104 µg/g of FW	Catechins > 69 µg/g of FW	-	Catechins > 109 µg/g of FW
Chlorogenic acid	62.6 µg/g of FW	60.2 µg/g of FW	-	1< µg/g of FW	466 µg/g of FW	-	72.4 µg/g of FW
Caffeic acid	17.6 µg/g of FW	150 µg/g of FW	-	5.1 µg/g of FW	5.8 µg/g of FW	-	41.3 µg/g of FW
Ferulic acid	21.7 µg/g of FW	38.5 µg/g	-	17.9 µg/g of FW	109 µg/g of FW	-	22.6 µg/g of FW
*p*-hydroxybenzoic acid	1.5 µg/g of FW	553 µg/g of FW	-	1.6 µg/g of FW	12.1 µg/g of FW	-	6.9 µg/g of FW
*p*-coumaric acid	21.1 µg/g of FW	97.3 µg/g of FW	-	23.9 µg/g of FW	32.4 µg/g of FW	-	16.6 µg/g of FW
***** TPC	124.5 µg/g of FW	899 µg/g of FW	-	48.5 µg/g of FW	625.3 µg/g of FW	-	159.8 µg/g of FW
****** Anthocyanidins	1294 µg/g of FW	112.9 µg/g of FW	-	2179 µg/g of FW	1850 µg/g of FW	-	996.3 µg/g of FW

FW—fresh weight; d—diameter; * TP—total phenolics; ** TA—total anthocyanins; *** AC—antioxidant capacity; **** F3 and F—Flavan-3-ol and Flavonol content; ***** TPC—Total phenolic acid content; ****** Anthocyanidins (cyanidin, delphinidin, malvidin, peonidin, and petunidin); ORAC—oxygen radical absorbance capacity.

Their indigenous distribution ranges chiefly in the western part of the North American continent, in the Rocky Mountains from the southwestern part of the Northwest Territories, northern Utah, and northern California, with disjunct populations in Michigan and Ontario to the east [152,153]. In Montana, the plant can also be found with the name globe huckleberry, with some plants regarded as *V. globular* in the Rocky Mountains. It distinguishes itself with remarkable ecological plasticity, reflected in the wide altitude range at which it can be found [148,150,151,153,154].

*V. membranaceum* is an upright, deciduous shrub standing from 0.3 to 2 m, forming small to vast clumps and seldom crown-forming. The fruit is spherical, about 1.5 cm in diameter, and sweet, with colours ranging from black to purple to red and, very rarely, white. Huckleberries require acidic and, usually but not imperatively, moist soils to become established, *V. membranaceum* being known for its difficulty in establishment.

### 4.2. Constituents

Mountain huckleberry is rich in aroma and flavour chemicals. Anthocyanins and polyphenolics are secondary plant metabolites. They are a varied group, with over 4000 flavonoids having been described. These compounds are essential to food quality through taste, appearance, and health benefits.

Through Lee’s et al. 2004 research, constituent nutrients of the *V. membranaceum* belonging to this group have been calculated. First, compound extraction (acetone and 70% aqueous acetone) occurred, after which the samples were partitioned (1:2 acetone chloroform). After evaporation and distillation, the samples were HPLC-filtered. This resulted in a pH of 2.6, 12.7 °Brix, total monomeric anthocyanin content of 167 mg of cyanidin 3-glucoside/100 g of frozen berries, and phenolic content of 617 mg of gallic acid/100 g of frozen berries. The major polyphenolics in *V. membranaceum* were neochlorogenic acid (3–*O*–caffeoylquinic acid, 26% of the total peak area measured at 320 nm) and a cinnamic acid derivative (peak 2, 21% of the total peak area measured at 320 nm). Minor polyphenolics were identified in the ethyl acetate fraction, such as protocatechuic acid, gallic acid, and epicatechin [155].

In another 2004 study by Lee et al., which analysed fresh huckleberry fruits, total monomeric anthocyanin content with a range of 101 to 360 mg/100 g and a content of total phenols membranaceum ranging from 367 to 1286 mg/100 g [145]. The first nutritional report of *V. membranaceum* dates from Kuhlein’s analysis of fresh berries in 1989. For 100 g, this fruit quantified moisture content of 86%, 0.6% protein, 0.5% fat, 13.1% carbs, 2% fibre, and 54 kcal. For minerals: 14 mg Ca, 17 mg P, 0.4 Na, 8 mg Mg, 0.2 mg Fe, 0.1 mg Zn, 0.1 mg Cu, 2.5 mg Mn, and 0.3 mg Sr. The most important vitamin is represented by 6.6 mg of ascorbate and 0.5 carotenes [156].

The 2004 study by Taruscio et al. highlighted the presence and content of flavonoids and phenolic acids and the antioxidant capacity of *V. membranaceum*. Thus, the total phenolics were found at 1.70 mg of GAE (gallic acid equivalents)/g of FW (fresh-frozen weight); total anthocyanins at 1.69 mg of C3G(cyanidin 3-glucoside)/g of FW; flavan–3–ol and flavonol content were measured for the catechins (catechin and epicatechin) level at over 240 µg/g of FW; the phenolic acids measurements: chlorogenic acid 62.6 µg/g of FW; caffeic acid 17.6 µg/g of FW; ferulic acid 21.7 µg/g of FW; *p*–hydroxybenzoic acid 1.5 µg/g of FW; *p*–coumaric acid 21.1 µg/g of FW; anthocyanidins at 1294 µg/g of FW; and with an antioxidant capacity at ORAC (oxygen radical absorbance capacity) 14.6 µmol of TE (Trolox equivalents)/g of FW and FRAP (ferric reducing ability of plasma) 30.2 µmol of TE/g of FW [149].

### 4.3. Ecological Value, Commerce, and Economy

Huckleberries have been important culturally and socioeconomically to the Native American population and an important food source for wild fauna. They represent forage for black and grizzly bears, with studies suggesting that these berries represent up to 10% of their diet, with range overlap between the two beings also observed, corroborating the indispensability of huckleberries from this bear’s diet and preparation for hibernation. Deer, elk, moose, foxes, small mammals, gulls, turkeys, and other birds consume parts of the shrub. Pollinators also benefit from this plant’s flowers; namely bees and bumblebees [148,154,157,158].

*V. membranaceum* is considered to have the best taste of all Western huckleberries and has a long, thousand-year-old harvesting history for personal and commercial use [146]. Even in modern times, the berries are harvested from wild, naturally occurring stands, such as fields and open-canopy, with the majority belonging to public U.S. lands [152].

The huckleberries have had an essential economic aspect in North America since becoming popular with the European colonists. Besides being a food source, they were used to obtain natural dyes or smoking mixtures. In the early 2000s, there was a sudden increase in demand. *V. membranaceum* has intense, easily identifiable flavours that fit into market niches. The berries are picked from wild stands between July and August and are eaten fresh, baked in pies, pancakes, syrups, candies, vinaigrettes and salad dressings, soft drinks and ales, made into jellies and jams, canned, or frozen. From huckleberry leaves, tea can also be produced. Likewise, there are shampoos, lotions, soaps, and other products made out of huckleberries. To match the demand, the University of Idaho has tried to produce cultivars for field cultivation and managed forest systems. In 2013, huckleberry products sales made over USD 1 M in revenue for Montana, with jellies, jams and preserves representing over 40% of the production. A big challenge for the commercial expansion of this fruit is characterized by the difficulty of its endemic environment, rough mountain sites, and lack of accessible infrastructure [142,143,146,148,151,152].

## 5. Economic Aspects of Using Waste and by-Products of Berries

The management of natural resources relies on the ecological consciousness of species and populations, along with the ethnic and social-economic features of a particular area [159,160]. Modern society has an ever-increasing requirement for using aromatic and medicinal natural resources [161,162]. The leaves are generally considered a by-product of berry-fruit production, even though they contain important content of phenolic compounds. Booth et al. (2012) analysed the cranberry leaf extract from a safety point of view. They found that the leaf is safe for consumption and is mainly composed of organic acids, simple sugars, catechins, flavonoids, fatty acid glycosides, proanthocyanidins, rosarin, and iridoids [163].

Meanwhile, bilberry leaves are primarily collected between May and June, throughout blooming, and they are mainly used as a traditional remedy. Conventionally, they are used particularly due to their anti-inflammatory, tonic, antiseptic, and astringent properties. They are also efficient in the treatment of UTIs, liver diseases, bladder stones, rheumatism, and other disorders [164]. A recent review article analysed the phenolic compounds of five *Vaccinium* species, namely blueberry, bilberry, bog bilberry, bearberry, and lingonberry. As concluded, only some studies have examined these aspects, even though they have numerous bioactive compounds [19]. The main phytochemicals identified within the leaves are arbutin, quercetin, and chlorogenic acid. Arbutin and its derivatives increase antimicrobial activity against Gram-negative bacteria (*Proteus vulgaris, E.coli, H. pylori*) [165]. Quercetin is efficient in avoiding multiple chronic diseases and can be mostly found in fruits with red leaves, along with kaempferol, proanthocyanidins, and hydroxycinnamic acids [166]. A high content of soluble phenolics are located in the leaves of these species, such as chlorogenic acid (55%) and flavonol glycosides (30%) [167].

From the fruits and leaves of berries, herbal tea can be produced, which displays essential antioxidant and antibacterial activity. The administration of these berry juices can hinder the development of atherosclerosis. After 12 weeks of consumption, the anthocyanins in these juices inhibited the aortic lipid accumulations between 79–96%, activated a diminished movement of hepatic antioxidant enzymes, and facilitated neuronal behaviour and functions [168]. Along with atherosclerosis, these compounds have also been applied to alleviate throat, mouth, and other infections, nausea, stomatitis, aphtha, diarrhoea, diabetes, inflammations, dysentery, and various cancer types [169,170].

Another by-product that is usually generated through the extraction of bioactive compounds for the production of pharmaceutical formulations is berry pomace. These by-products are still full of valuable compounds, bioactive phytochemicals and nutrients. To reduce these “wastes”, the extraction should be effectuated with food-grade solvents. Afterwards, as a recent study demonstrated, good functional ingredients can be formulated from these by-products with multistep high-pressure fractionation [171].

As several studies highlight, using the by-products of berry fruit production presents an essential economic aspect through the reduction of wastes and valorification of them within functional food products, nutraceuticals, or supplements. These wastes are also rich or even richer in phenolic compounds, flavonoids, and anthocyanins than fruits.

## 6. Conclusions and Perspectives

Parts of *Vaccinium* species, such as leaves, stems, and especially fruits, are widely utilized in functional food, in various traditional medicinal products, cosmetics, and even in ornamental product generation. Their abundant bioactive compounds, antidiabetic, anti-inflammatory, anti-carcinogenic, antioxidant, and other health-related effects, are extensively used to treat multiple diseases.

The *Vaccinium* genus found worldwide, especially those belonging to cranberry, blueberry (bilberry), and huckleberry species, have been analysed regarding their distribution, history, uses, health benefits, and chemical profile. As can be observed, the huckleberry species still needs some further studies into applicability.

Additionally, an important aspect should be the focus on the by-products, such as the leaves of most cultured berries worldwide. The bioavailability of the identified polyphenols and their mechanism of action should be better investigated. Additionally, the integration of bioactive compounds extracted from leaves and their introduction in nutraceuticals or functional foods should be further evaluated.

## Figures and Tables

**Figure 1 molecules-28-01533-f001:**
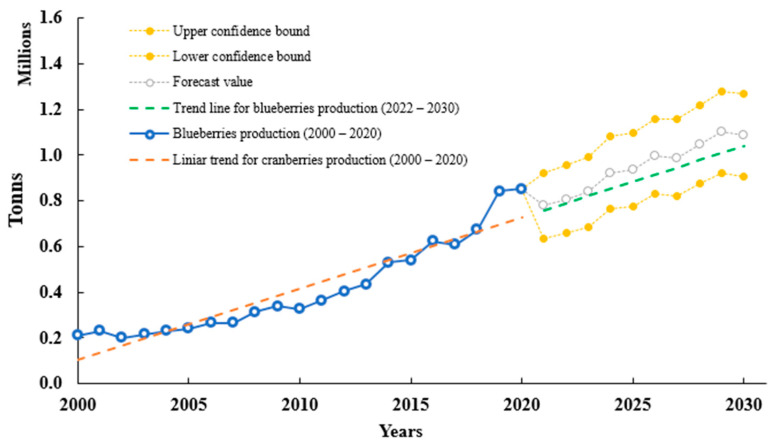
Evolution of world blueberry production over time. According to FAO, global blueberry production (tons) from 2000 until 2030 and TREND production until 2030. The trendline was calculated considering blueberry production every year from 2000 to 2030. The TREND function returns values along with a linear trend. It fits a straight line (using the method of least squares) to the arrays known_ys and known_xs. TREND returns the y-values along that line for the array of new_xs that you specify.

**Figure 2 molecules-28-01533-f002:**
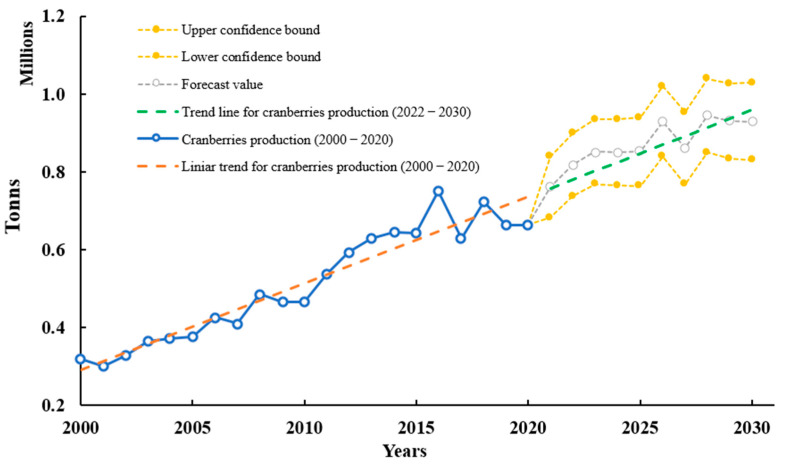
Evolution of world cranberry production over time. According to FAO, global cranberry production (tons) from 2000 until 2030 and TREND production until 2030. The trendline was calculated considering cranberry production every year from 2000–2020. The TREND function returns values along with a linear trend. It fits a straight line (using the least squares method) to the arrays known_ys and known_xs. TREND returns the y–values along that line for the array of new_xs that you specify.

**Figure 3 molecules-28-01533-f003:**
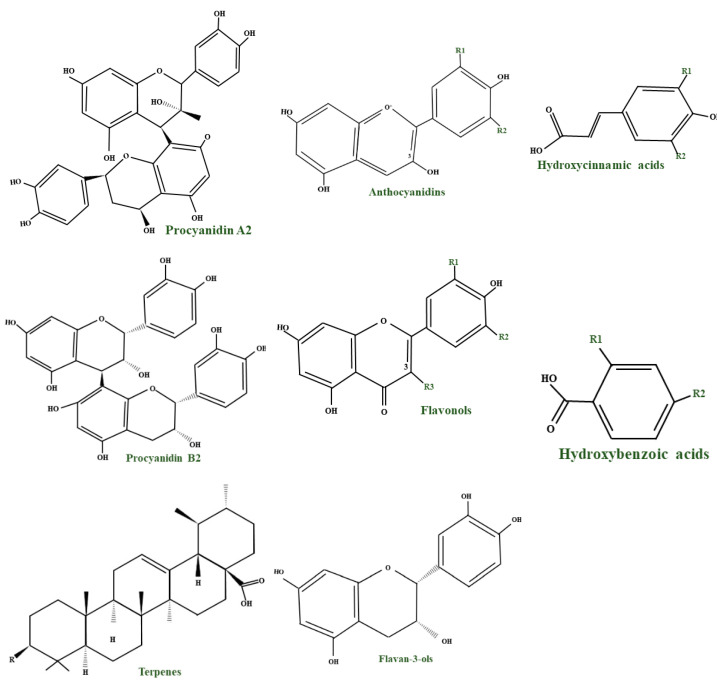
Bioactive compounds found in cranberry; figure adapted from [115].

**Table 1 molecules-28-01533-t001:** Phenolic compounds identified in aerial parts of *Vaccinium* plants.

Class	Phenolic Compounds		Chemical Structures (Main Compound)
	Leaves	Stems	
Catechins	(+)-catechin—R1 = H (–)-epicatechin—R1 = H+ gallocatechin—R1 = OH epigallocatechin—R1 = OH		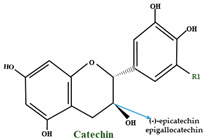
Cinchonains	cinchonains I cinchonains II		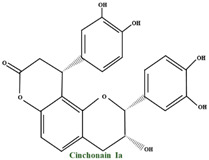
Phenolic acids	3,4–dihydroxybenzoic *p*–coumaroyl quinic acid isomers *p*–coumaroyl malonic acid *p*–coumaroyl derivatives *p*–coumaroyl glucose coumaroyl iridoid		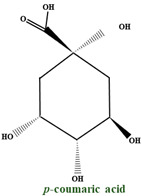
	*p*–coumaric acid	-	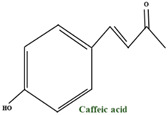
	feruloyl quinic acid isomer	-	
	caffeoyl quinic acid isomers	-	
	caffeic acid ethyl ester	-	
	caffeic acid hexoside	-	
Proantho- cyanidins	B–type dimer B–type trimer B–type tetramer B–type pentamer A–type dimer A–type trimer procyanidin A2 procyanidin B1 procyanidin B2		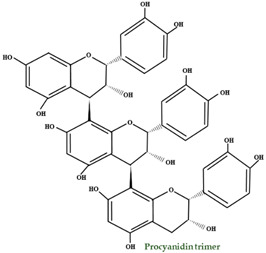
Flavonols	quercetin–3–*O*–(4”–HMG)–α–rhamnoside quercetin–3–*O*–galactoside quercetin–3–*O*–glucoside quercetin–3–*O*–rutinoside quercetin–3–*O*-α–rhamnoside quercetin–3–*O*–arabinoside quercetin–3–*O*–glucuronide		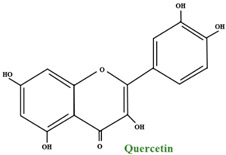
	kaempferol	-	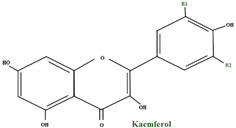
	kaempferol–3–glucuronide	-	
	kaempferol–hexoside	-	
	kaempferol–*O*–pentoside	-	
	kaempferol–(HMG)–rhamnoside	-	
Lignans	-	Lyoniside (9–*O*–β–D–xylopyranosyl(+)lyoniresinol)	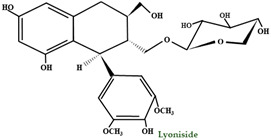

**Table 2 molecules-28-01533-t002:** Traditional uses of *V. myrtillus*.

Traditional Uses	Used Parts	Administration	References
Fevers and coughs	Fruits	50–200 mg/kg	[63]
Respiratory inflammations	Leaves/fruits		[58,64]
Antidiabetic	Leaves	10 mg	[65]
Anti-inflammatory	Leaves Fruits	4 mg/mL 100 ng/mL	[66] [67]
α-glucosidase activity	Fruits	20 μg GAE/mL ME 55 μg GAE/mL AE	[61]
Neuroprotective effects	Fruits	100 mg/kg	[68]
Digestive Urinary tract disorders	Fruits	250 mg GAE/L	[22]
Eye inflammation	Fruits	50 mg/mL	[69]
Ulcerative colitis	Fruits	95 g DW	[70]
Hypertension Weight gain reduction	Fruits	3% (*w*/*w*) 10% (*w*/*w*)	[71]
Sun Protection Factor (SPF)	Fruits	2.84 g/100 gDW	[58]
Diabetic retinopathy	Leaves/fruits	100 mg/kg	[72]
Antiseptic, astringent, tonic	Fruits	28.3 μg CYA-3-GLU equivalents/mL CM	[73]
Angiotensin-Converting Enzyme Activity	Fruits	0.1 mg/mL	[74]

GAE—gallic acid equivalents; DW—dry weight; CM—culture medium; ME—methanol extract; AE—aqueous extract.

**Table 3 molecules-28-01533-t003:** Content of macro- and micro-elements in *V. macrocaropn* and *V.oxycoccos*. (adapted from [107,108,109]).

*V. macrocarpon*		*V. oxycoccos*	Method
**Macroelements, mg/100 g fw**			
Magnesium	5.2–9.1	7.8–9.1	AAS (Perkin Elmer AAnalyst 700, acetylene-air flame)
Sulfur	5.2–14.3	6.5–18.2	Turbidimetry
Potassium	52.0–98.8	78.0–93.6	FP (Jenwey PFP7, airpropane butane flame)
Calcium	7.8–14.3	9.1–18.2	AAS (Perkin Elmer AAnalyst 700, acetylene-air flame)
Phosphorus	6.5–11.7	6.5–7.8	Colourimetry
Nitrogen	10.4–65.0	13.0–78.0	Colourimetry
**Microelements, mg/100 g fw**			
Iron	0.22–1.17	0.33–0.42	AAS (Perkin Elmer AAnalyst 700, acetylene-air flame)
Manganese	0.06–0.57	2.18–3.95	AAS (Perkin Elmer AAnalyst 700, acetylene-air flame)
Zinc	0.07–0.17	0.14–0.19	AAS (Perkin Elmer AAnalyst 700, acetylene-air flame)
Molybdenum	0.01–0.02	0.01–0.02	Colourimetry
Copper	0.04–0.08	0.06–0.08	AAS (Perkin Elmer AAnalyst 700, acetylene-air flame)
Boron	0.03–0.12	0.10–0.17	Colourimetry

AAS—atomic absorption spectrophotometer; FP—flame photometer.

**Table 5 molecules-28-01533-t005:** Prevention of different types of diseases with cranberry consumption and mechanisms.

Type of Disease	Mechanism	References
Gastric cancer	A-type procyanidins in cranberry ↓ the adhesion of *H. pylori* on human gastric mucus and have an inhibitory effect	[125,126]
Obesity	Cranberry ↓ lipid accretion by lowering mRNA level of some genes associated with fatty acid-binding protein, lipoprotein lipase, fatty acid synthase, hormone-sensitive lipase, and perilipin	[127,128,129]
Type 2 diabetes	Quercitin ↓ gastric assimilation of glucose and with myricetin can ↓ glucose transporter type 4 mediated glucose assimilation	[130,131]
Urinary tract inflammation	A-type PACs ↓ adhesion of P-fimbriated uropathogenic *E. coli* to uroepithelial cells	[114,116,132]
Periodontitis	Cranberry non-dialysable material ↓ proliferation of *P.gingivalis*, *T.forsythia*, and *T.denticola* in periodontal pockets A-type cranberry PACs ↓ production of metalloproteinases	[124,133,134]
Anti-inflammatory	Quercetin ↓ of the nuclear factor-kappa B (NF-κB) pathway, NDM lower lipopolysaccharide-induced inflammatory cytokine production	[135]
Nonalcoholic fatty liver disease	In vitro study > cranberry diet 3 months > alanine reduction aminotransferase and insulin, + lipid profile effect, insulin resistance and hepatic steatosis in NAFLD patients	[136]
Microbial	Cranberry extracts ↓ on pathogenic bacteria: *Listeria monocytogenes* (ATCC 19117), *B. cereus* (ATCC 10876), *B. subtilis* (ATCC 6633), *M. luteus* (ATCC 9341), *E. faecalis* (ATCC 29212), *S. aureus* (ATCC 25923) and Gram-negative *E. coli* (ATCC25922), *Enterobacter aerogenes* (ATCC 13048), *Slm. typhimurium* (ATCC 14028), and *Slm. agona* bacteria	[122,123]
Cardiovascular	The expression of inflammatory genes suited for cardiovascular diseases is ↓ by resveratrol (polyphenol present in cranberry juice) by inflecting the NF-kB and JAK STAT3 pathways in cultured cells	[137,138]
Breast and colon cancer	Quercitin and proanthocyanidin inhibited the expansion of MCF-7 human breast adenocarcinoma and HT–29 human colon adenocarcinoma	[106,118]
Leukaemia and lung cancer	Ursolic acid can inhibit the growth of some leukaemia cell lines and A-549 human lung carcinoma	[106,139]

Note: JAK—(Janus kinase inhibitor); NF-kb—Nuclear factor kappa-light-chain-enhancer of activated B cells; STAT3—signal transducer and activator of transcription 3; PACs—proanthocyanidins.

## Data Availability

Not applicable.

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
