# Peer review of "Vaccinium* Species (Ericaceae): Phytochemistry and Biological Properties of Medicinal Plants"

_molecules, 2023, doi:10.3390/molecules28041533_

Round 1
Reviewer 1 Report
Review:
The review paper entitled ‘Vaccinium Species (Ericaceae): Phytochemistry and Biological Properties of Medicinal Plants’ by Martău’ et al. is a very poorly constructed review paper with a lot of redundant text and without focus. Most of the text can be omitted from the paper. The authors should concentrate on the phytochemistry rather than the history and nomenclature of the plants. In addition, the chemical scaffolds presented in the text are not uniform and should be redrawn with the same sizes.
The usage of the English language is poor, and the authors should take the help of some native speakers to recheck their paper. The paper needs to be revised thoroughly.
Some of the comments are as below:
325-330: Poor presentation, with incomplete sentences, and a lot of typos
329: hexan or hexane?
A lot of redundant text. Should try efforts to remove the redundant substantial text.
343-344: powerful content of antioxidants. What does it mean? It’s not scientific at all.
352: almost attenuated. What does ‘almost’ mean?
373: constantly adequately?
501 and 509: please provide recent data. Data from 2007 is outdated.
507-508. Please rewrite.
507-518: The paragraph describes the production. Nothing about the biological properties, thus the section can be omitted including the figure. This makes no sense at all.
543-544: Please rewrite.
569: at women??
621-624: please rephrase
656-660,662, 666: please italicize scientific names
Section 4: History, dimensions and growth, nomenclature, etc are not necessary for the phytochemistry paper. Please remove those sections.
Author Response
Dear Reviewer #1,
We would like to take this opportunity to express our sincere thanks for identifying areas of our manuscript that needed corrections or modifications. The suggestions and recommendations helped us to improve the quality and the content of the paper. We appreciate the time dedicated to read and review our manuscript.
We have revised the manuscript by modifying all the mentioned sections/aspects, based on your comments and recommendations. Accordingly, below we have responded point-by-point to each comment.
|
Reviewer 1 - Specific Comments: |
Revision |
|
The review paper entitled ‘Vaccinium Species (Ericaceae): Phytochemistry and Biological Properties of Medicinal Plants’ by Martău’ et al. is a very poorly constructed review paper with a lot of redundant text and without focus. Most of the text can be omitted from the paper. The authors should concentrate on the phytochemistry rather than the history and nomenclature of the plants. In addition, the chemical scaffolds presented in the text are not uniform and should be redrawn with the same sizes. |
We do appreciate your feedback regarding the whole manuscript. We have clarified and improved all the mentioned aspects, and we tried to concentrate mostly on phytochemistry. |
|
The usage of the English language is poor, and the authors should take the help of some native speakers to recheck their paper. The paper needs to be revised thoroughly.
|
We appreciate your recommendation, and in accordance, the manuscript was revised thoroughly by a native English speaker. |
|
325-330: Poor presentation, with incomplete sentences, and a lot of typos |
We kindly thank you for the relevance of this suggestion, indeed there were several errors. We corrected it as follows: “BE and its fractions, principally rich in β–sitosterol, containing 280 mg/g and 70 mg/g, exhibited high inhibition potential against S. aureus strains. Therefore, chloroform and ethyl acetate fractions had 20 and 15-mm inhibition zones. The chloroform and ethyl fractions were also characterized by an inhibitory effect against other gram-positive bacte-ria, counting S. pyogenes and S. epidermidis. All tested bilberry fractions showed an inhibi-tory effect against S. pyogenes, except the most non-polar hexane fraction. The tested berry lipid fractions had an inhibitory effect against only one gram-negative bacteria, E. coli. In-hibition zones of 11 mm were presented by the bilberry ethyl acetate/methanol and meth-anol fraction [49].” |
|
329: hexan or hexane? |
Thank you very much for the suggestion and observation regarding the correctness of hexane. We have corrected accordingly. |
|
A lot of redundant text. Should try efforts to remove the redundant substantial text.
|
Kindly thank you for this observation as it highly improves the quality of the manuscript. We have removed a lot of redundant text, and tried to improve the focus of the text.
|
|
343-344: powerful content of antioxidants. What does it mean? It’s not scientific at all.
|
As kindly suggested, we have corrected this section as follows: “The anti-inflammatory activities of bilberry fruits and their extracts are based on their high anthocyanin content. Mirtoselect is a commercial BE, standardized to a concentra-tion of anthocyanins of 40%. This extract showed a complex anti-inflammatory response in lipopolysaccharide-activated macrophages.” |
|
352: almost attenuated. What does ‘almost’ mean?
|
Thank you very much for the suggestion and observation. We corrected this phrase as following: “The feedback of the human THP–1 monocytic cell after the treatment with anthocya-nin-rich bilberry extract to inflammatory stimulation was more diverse. BE attenuated the IFN–γ–induced signal protein activation, cytokine secretion, and pro-inflammatory gene expression since it increased TNF–α–induced reactions [63].” |
|
373: constantly adequately?
|
Thank you for this observation, we have corrected and revised this sentence: “Inflammatory bowel disease, a functional gastrointestinal disorder [66] often lacks proper treatment, having several adverse effects. The necessity to find a suitable and effec-tive treatment is reasonable. Evidence shows that a dysregulated immune response refer-ring to intestinal microbiota causes the inflammation.” |
|
501 and 509: please provide recent data. Data from 2007 is outdated.
|
Thank you very much for the suggestion and observation, but as also mentioned at another correction the growth area is not “necessary for the phytochemistry paper”. Consequently, this section has been removed. |
|
507-508. Please rewrite.
|
Kindly thank the reviewer for this observation; we rewrote this section: “Cranberry (Vaccinium sect. Oxycoccus) is one of the most cultivated fruits in the U.S., with a global production of over 663.000 tons in 2020; this had a 108% increase from 2000 to 2020 (FAO). Based on the production and consumption of cranberry, this will also pre-sent a constant growth trend, with an increase of 44,4% (958107 tons) by 2030.” |
|
507-518: The paragraph describes the production. Nothing about the biological properties, thus the section can be omitted including the figure. This makes no sense at all.
|
Highly appreciate the suggestion. We have modified and reduced many sections, but we think that the total production and the trendline of this is important, as itt grows due to the numerous health effects (biological properties). |
|
543-544: Please rewrite.
|
Thank you, we have rephrased this sentence: “Cranberries' therapeutic potential, in the case of both species V. macrocarpon and V.oxycoccos, comes from their antioxidant effect, which is given by many phytochemicals.” |
|
569: at women??
|
Kindly thank you for the relevance of this comment, we have corrected this sentence also: “Besides, they improve insulin secretion (which can help people with type two diabetes), and help especially women against urinary tract infections which are one of the most common bacterial infection [102].” |
|
621-624: please rephrase
|
Thank you for this observation. We have corrected the sentence: “Wu et al. (2020) identified in cranberry extract significant quantities of ursolic acid (372.97 mg/g), oleanolic acid (79.16 mg/g), β–sitosterol and stigmasterol (107.83 mg/g).” |
|
656-660,662, 666: please italicize scientific names
|
Thank you for this comment. We have corrected the scientific names to italics. |
|
Section 4: History, dimensions and growth, nomenclature, etc are not necessary for the phytochemistry paper. Please remove those sections. |
Kindly thank you for the relevance of this comment, we have removed several large sections regarding thesee aspects, only some minor parts remained. |
We would also like to thank you for allowing us to resubmit a revised copy of the manuscript.
Sincerely Yours,
Authors

Reviewer 2 Report
The paper is within the journal topic and presents some interesting Information. However the organization seems inappropriate since the Vaccinium Species were presented separately and the review organization does not regroups the related information: Blueberry(bilberry), Cranberry, Vaccinium membranaceum Huckleberry . This organization is confusing since the different species as presented in the Manuscript may be published in separate reviews.
· Line 96-99: please update data and statistic if there is any recent data
· Line 124: why the sugar content is reported as mmol/kg and not mg/kg. the last will make comparison easier.
· Table1: title, please revise to get a correct sentence “table adapted after [23,30,39]. »
· Table 2: please can you add in the table further information on administration of Vaccinium myrtillus, (used dose and method). Same remark for table4.
· The paper needs should be revised to meet English standards
Author Response
Dear Reviewer #2,
We would like to take this opportunity to express our sincere thanks for identifying areas of our manuscript that needed corrections or modification. The suggestions and recommendations helped us to improve the quality and the content of the paper. We appreciate the time dedicated to read and review our manuscript.
We have revised the manuscript by modifying all the mentioned sections/aspects, based on your comments and recommendations. Accordingly, below we have responded point-by-point to each comment. To improve the quality of English, a native English-speaking colleague within our university carefully checked the entire manuscript.
|
Reviewer 2 - Specific Comments: |
Revision |
|
The paper is within the journal topic and presents some interesting Information. However the organization seems inappropriate since the Vaccinium Species were presented separately and the review organization does not regroups the related information: Blueberry(bilberry), Cranberry, Vaccinium membranaceum Huckleberry . This organization is confusing since the different species as presented in the Manuscript may be published in separate reviews. |
We thank the reviewer for this positive appraisal of the manuscript based on the presented information. We also appreciate the comments received from the reviewer, which were very helpful in improving the manuscript. Based on the comments, we reduced the presented information and the organisation was also rethought. |
|
· Line 96-99: please update data and statistic if there is any recent data
|
We appreciate your recommendation, and in accordance, we have updated the data and statistics: “In the last five years, the production of blueberry was mainly focused in North America [34], with over 80% of world production [33], and with a production that reached over 1 million (M) tons in 2021 [32,35].” |
|
· Line 124: why the sugar content is reported as mmol/kg and not mg/kg. the last will make comparison easier.
|
Based on the reviewers recommendation we have reported the data as g/kg: “Bilberries incorporate high amounts of sugars (68.5-205.3 g/kg), organic acids (5.7-14.2 g/kg), and a mixture of phenolic compounds with multiple biological effects.” |
|
· Table1: title, please revise to get a correct sentence “table adapted after [23,30,39]. »
|
Thank you for this observation, the title of the table has been revised “Table 1. Phenolic compounds identified in aerial parts of Vaccinium plants.” and integrated in the text: “Phenolic compounds represent the primary group of phytochemicals found in ber-ries. They acquire one or more aromatic rings with hydroxyl groups, and their construc-tion may range from a simple phenolic molecule to a complex high-molecular-mass polymer, as can be seen in table 1, that is adapted after [23,30,39].” |
|
· Table 2: please can you add in the table further information on administration of Vaccinium myrtillus, (used dose and method). Same remark for table4. |
We appreciate the pertinent observation and as follows we have introduced further informations regarding the administration of Vaccinium myrtillus extract, in case of different diseases. |
|
· The paper needs should be revised to meet English standards |
Kindly thank you for the suggestion. To improve the quality of English, a native English-speaking colleague within our university carefully checked the entire manuscript. |
We would also like to thank you for allowing us to resubmit a revised copy of the manuscript.
Sincerely Yours,
Authors

Round 2
Reviewer 1 Report
Dear Authors,
I am writing to extend my warm congratulations on the revisions you have made to your manuscript. I have reviewed the updated version and I am pleased to inform you that you have addressed all the comments that were provided.
I appreciate the hard work and dedication you have put into revising the manuscript. The changes you have made have significantly improved the overall quality and clarity of the manuscript.
I am confident that your manuscript will make a valuable contribution to the field of plant pharmacology. I am glad that you have taken the time to incorporate feedback and make the manuscript stronger.
I wish you all the best for the publication of your work.
Sincerely,
Author Response
Dear Reviewer #1, we sincerely thank you for taking the time to revise our manuscript, and for the nice comments that you addressed to us.
Reviewer 2 Report
The paper is well revised according to the previous comments. I have no further comments.
Author Response
Dear Reviewer #2, the authors sincerely thank you for taking the time to revise the manuscript entitled 'Vaccinium Species (Ericaceae): Phytochemistry and Biological Properties of Medicinal Plants', and for the nice feedback sent to them.